# Oral Mucositis Induced by Chemoradiotherapy in Head and Neck Cancer—A Short Review about the Therapeutic Management and the Benefits of Bee Honey

**DOI:** 10.3390/medicina58060751

**Published:** 2022-05-31

**Authors:** Daniela Jicman (Stan), Mihaela Ionela Sârbu, Silvia Fotea, Alexandru Nechifor, Gabriela Bălan, Mihaela Anghele, Claudiu Ionuț Vasile, Elena Niculeț, Nicolae Sârbu, Laura-Florentina Rebegea, Alin Laurențiu Tatu

**Affiliations:** 1Department of Otorhinolaryngology, “Sfântul Apostol Andrei” Emergency Clinical Hospital, 800578 Galați, Romania; dana.corche@yahoo.com; 2Biomedical Doctoral School, Faculty of Medicine and Pharmacy, “Dunărea de Jos” University, 800010 Galați, Romania; anghele.mihaela@yahoo.com; 3Clinical Medical Department, Faculty of Medicine and Pharmacy, “Dunărea de Jos” University, 800008 Galați, Romania; silvia_ghimpu@yahoo.com (S.F.); alexandrunechiformed@yahoo.com (A.N.); gabrielamedicine@yahoo.com (G.B.); claudiumvasile@yahoo.com (C.I.V.); helena_badiu@yahoo.com (E.N.); 4Multidisciplinary Integrated Center of Dermatological Interface Research (MIC-DIR), “Dunărea de Jos” University, 800008 Galați, Romania; 5Gastroenterology Department, “Sf. Apostol Andrei” County Emergency Clinical Hospital, 800578 Galati, Romania; 6Clinical Emergency Hospital “Sfantul Apostol Andrei”, 800578 Galati, Romania; 7Department of Morphological and Functional Sciences, Faculty of Medicine and Pharmacy, “Dunărea de Jos” University of Galați, 800010 Galati, Romania; 8Faculty of Medicine and Pharmacy, “Dunărea de Jos” University, 800008 Galati, Romania; dnsarbu@gmail.com; 9Radiotherapy Department, “Sfântul Apostol Andrei” Emergency Clinical Hospital, 800578 Galați, Romania; 10Research Center in the Field of Medical and Pharmaceutical Sciences, ReFORM-UDJ, 800010 Galati, Romania; dralin_tatu@yahoo.com; 11Dermatology Department, “Sfânta Cuvioasă Parascheva” Clinical Hospital of Infectious Diseases, 800179 Galati, Romania

**Keywords:** oral mucositis, honey, head and neck cancer, chemotherapy, radiation therapy

## Abstract

*Background and Objectives*: Oral mucositis, a severe non-hematological complication, can be induced by chemoradiotherapy. It is associated with severe local dysfunction, severely affecting the patient’s quality of life; it increases the risk of oral infections and interrupts oncological treatment, thus prolonging the duration and cost of hospitalization. Besides all of the agents used in the prevention and treatment of oral mucositis induced by oncological treatment, can there be found an easier one to administer, with an effective preparation, high addressability, both for adults and paediatric patients, without side effects, and at the same time cheap and easy to purchase? The aim of the present paper is to demonstrate the existence of this product, which is available to everyone, having multiple benefits. *Materials and Methods*: For the purpose of writing this article, materials were searched in electronic databases in between 2019 and 2021, taking into consideration papers where authors have demonstrated the effectiveness of this product through its topical or systemic use. *Results*: Numerous studies have highlighted the benefits of honey on oral mucositis. Through its analgesic, anti-inflammatory, anti-cancerous and antibacterial action, honey has proved to have a major impact on the patient’s quality of life and nutritional status by promoting tissue epithelialization and healing of the chemoradiotherapy-induced lesions. *Conclusions*: Superior to many natural agents, bee honey can be successfully used in both preventing and treating oral mucositis. There are currently numerous studies supporting and recommending the use of bee honey in the management of this oncological toxicity.

## 1. Introduction

Currently epidemiological data support that head and neck tumors are a public health problem due to their increasing incidence, prevalence and high mortality. Head and neck cancer (HNC) accounts for more than 550,000 cases and 380,000 deaths annually, worldwide [1].

Head and neck oncology faces real challenges as these types of tumors have serious repercussions on the patient’s quality of life. They affect various areas that directly interfere with the patient’s everyday life, such as: speech, taste, ability to chew, swallow, breathe, facial bone changes, dental mobility, local or remote vascular-nerve functionality, physical appearance; these types of cancers can also have a profound, long-lasting psychological impact, sometimes with extensive recovery processes. Therefore the treatment of Ear, Nose & Throat (ENT) cancers is extremely complex, multimodal and requires, on one hand, the involvement of a multidisciplinary therapeutic team, which must include medical oncologists, radiation therapists, pathologists, dentists, nutritionists, and on the other hand, the availability of certain technologies and techniques for radiotherapy (3D conformal 3DCRT or intensity modulated, IMRT). In addition to surgery and radiotherapy, chemotherapy and immunotherapy have undergone major developments, based on new and effective clinical trials but with more or less manageable toxicities [2,3].

Of all chemoradiotherapeutic toxicities, OM seems to be difficult to manage and despite many studies, there is no unanimously accepted protocol by clinicians today [4].

Considered one of the most severe non-hematological complications, OM seriously affects the patient’s quality of life. It is associated with severe local dysfunctions, increases the risk of oral infections, may interfere with oncological treatment prolonging the duration and cost of hospitalization [5].

From a radiobiological point of view, the oral mucosa is one of the acute-response tissues whose lesions may be reversible and frequently occur during irradiation or after treatment completion [6].

However, radiotherapy has an adverse influence on other mucosae and the inflammatory and indurated cutaneous changes associated with mucositis (as an expression of the effects of radiotherapy) must be clinically dissociated from other dermatological manifestations, similar in clinical expression: atrophic lichen sclerosus, morphea or lichen planus [7].

In order to prevent and treat chemoradiotherapy-induced OM, a variety of natural and synthetic substances are currently used. In addition to the recognized and marketed recommendations, more and more emphasis is being placed on natural products in view of their minimal adverse effects [4].

One of the best known natural agents is honey, which can be defined as a heterogeneous mixture of substances such as proteins, sugars, which come from the nectar of flowers and glandular secretions produced by bees. Honey is an extremely complex product and can be considered both a plant and an animal product [8,9]. Honey contains more than 200 different natural compounds, grouped into macro and micronutrients, depending on the type of bees, natural floral source, environmental factors and processing methods. Among the compounds that make up honey are: sugar, proteins, enzymes, minerals, vitamins, amino acids and a wide range of polyphenols. These compounds give honey its color, taste, viscosity and various therapeutic properties. Various scientific studies have shown that honey has numerous benefits. Thanks to flavonides and phenolic acids, honey has important antimicrobial activity. Moreover, it has antiviral, antioxidant, anti-inflammatory and antineoplastic effects [10]. These properties can be attributed to physicochemical characteristics such as high osmolarity and low pH, due to the presence of organic acids; the concomitant effects of the antioxidant and antimicrobial properties of honey, together with the anti-inflammatory properties, produce a healing effect on lesions. Honey is a good preventive agent against bacterial effects because its physicochemical properties provide an environment that is not conducive to bacterial proliferation, thus inhibiting the inflammation process [11].

As it can be seen, honey, as a natural agent has a broad-spectrum activity comparable to other systemic anti-inflammatory agents, e.g., dexamethasone, which due to its anti-inflammatory and other effects can be used in dermatology, oncology, surgery, etc. [12]. Honey also stimulates the immune system by producing antibodies [13]. Research has established that honey has a strong impact on the proliferation of B and T lymphocytes, thereby activating macrophages. This process results in the inhibition of the inflammatory process by inhibiting the cyclooxygenase pathway, which is recognized as the main pathway of the inflammatory process. Honey also stimulates the process of granulation, with angiogenesis, epithelialization and fibroblast proliferation. As established, the mechanisms involved in the antibacterial activity of honey are based on certain enzymes, phytochemicals, low pH, certain peptides and high osmolarity [8].

## 2. Materials and Methods

For the purpose of writing this article, materials were searched in electronic databases such as PubMed or Google Scholar, between the years 2019–2021, identifying 395 results. Using the keywords “oral mucositis; honey; head and neck cancer; chemotherapy; radiation therapy” all screened references were viewed and entered into electronic files. From these, we used studies and articles in English, as full-text, review articles and meta-analyses with the subject of: use of bee honey for preventing and treating head and neck cancer, chemoradiotherapy-induced OM. Articles targeting cancer patients treated only with radiotherapy were excluded. After eliminating duplicates, incomplete articles, dissertation papers and case reports, 3 studies, one of which was a murine model, 4 review articles, 4 meta-analyses and 1 Bayesian analysis were regarded as useful for discussion. Also, articles supporting the beneficial results of honey in the treatment of OM in pediatric patients from intensive care units and articles on the use of honey in viral-induced stomatitis have been mentioned. Exceptionally, studies supporting and recommending the use of honey in patients with colon cancer requiring higher doses of chemotherapy have also been mentioned. Thus, current recommendations based on large studies, also target honey for this purpose due to its properties.

## 3. Results

### 3.1. Incidence and Pathophysiology of Chemoradiotherapy-Induced Oral Mucositis

OM is one of the most common and painful side effects of chemoradiotherapy. It is defined as an inflammatory lesion of the oral mucosa, manifested by atrophy, edema, erythema, ulceration and pseudomembrane formation [8]. OM generally occurs in patients undergoing head and neck radiotherapy and it affects 75–100% of patients receiving high-dose chemotherapy; it also develops in patients with haematopoietic stem cell transplantation. It affects 20–40% of patients receiving conventional chemotherapy [14]. Different anti-neoplastic agents have been taken into consideration by different standardized protocols, but registered high toxicities. Chemotherapy in head and neck cancers is administered in adjuvant and neoadjuvant regimens, as well as concomitantly with radiotherapy. The main chemotherapies used in the treatment of head and neck neoplasms are: alkylating agents (carboplatin, cisplatin), antimetabolites (5-fluorouracil, gemcitabine, methotrexate), antitumor antibiotics-anthracyclines or non-anthracyclines (doxorubicin, bleomycin), mitotic inhibitors (docetaxel, paclitaxel), immunotherapeutics, targeted therapies and hormone therapy [15]. These chemotherapies are administered as monotherapy or in combination, increasing the risk of side-effects [16]. For example, induction chemotherapy has seen unprecedented growth with the advent of cisplatin administered with paclitaxel or docetaxel and cisplatin with gemcitabine. These drugs represent the greatest risk factor in the development of OM [16,17].

Administered alone or in various combinations, chemotherapy with radiotherapy induces oral mucositis and other toxicities [18]. The installation of OM is a limiting factor in the administration of chemotherapy concurrently with radiotherapy in head and neck neoplasms. 

The occurrence of OM is related to the action of cytostatic medication and radiotherapy by direct mechanisms, through the occurrence of apoptosis, and indirect mechanisms, through which proinflammatory mediators such as tissue necrosis factor, interleukin 1 beta and 6 are released, with the concomitant decrease of other anti-inflammatory cytokines, such as: interleukin 10 or transforming growth factor beta [19].

The oral mucosa is particularly sensitive to anticancer treatment [14]. Chemoradiotherapy-induced OM has direct DNA strand breaks localized in the basal epithelium, thus resulting in the release of reactive oxygen species, causing direct damage to mucosal cells [20].

Combination chemotherapy or high-dose chemotherapy increases the risk of OM, which is considered acute in nature, manifesting with ulceration of the oral mucosa which occurs in the first week after the initiation of treatment, and which heals within 3 weeks after the end of treatment. In this case, mucositis may also appear in the context of severe leukopenia. Radiotherapy-induced mucositis occurs due to the necrotic and inflammatory effects of radiation on the oral mucosa, and it is considered as a chronicevent, with ulceration appearing around week 2 of the 6–7 week cycle and healing almost spontaneously 3–4 weeks after treatment [21].

The difference between chemotherapy-induced OM and radiotherapy-induced oral mucositis can be seen in the manner and timing of the development and healing of the lesions; mucositis induced by targeted therapies is different from the two types listed above, and is manifested clinically by ulcerations that appear similar to aphthous stomatitis [22].

### 3.2. Risk Factors and Pathogenesis of Oral Mucositis

The risk factors for chemoradiotherapy-induced OM are, on one hand, patient-dependent and on the other hand, associated with antineoplastic treatment; the patient-associated factors include: age, body mass index, nutritional status, environment and oral hygiene before or during chemotherapy, use of tobacco, alcohol intake during treatment (which may aggravate OM lesions), pre-existaing and concomitant chronic conditions, gender and genetic predisposition; those related to treatment are: tumor location, type, dose, combination of chemotherapy with radiotherapy, irradiation technique, route of administration, even the tumor itself [21,22,23]. 

Regarding the irradiation technique, the old techniques, i.e., 2D, produced OM in a higher and more severe percentage compared to modern techniques 3DCRT, IMRT.

However, OM is frequently encountered especially in sequential or concurrent chemotherapy and radiotherapy in head and neck tumors. The pathogenesis of OM refers to the 5 phases recognized today by clinicians, namely: initiation of tissue damage, presence of inflammation by generating messenger signals, signaling and signal amplification, ulceration, inflammation and scarring, respectively (see Figure 1) [17,24].

Initially, OM is clinically manifested by a mucosal rash, with no other lesions or ulcerations being present; the patient only feels a burning sensation. These early symptoms appear 3–5 days after the start of chemotherapy, peaking at 7–14 days after therapy initiation and extending up to 3 weeks [25]. However, these manifestations can progress to severe stages, with the development of deep and extremely painful ulcerative lesions that make it impossible to hydrate or even to speak and swallow. These ulcerative lesions found in this stage are different from those of aphtous stomatitis and are differentiated from them by the lack of a peripheral ring of erythema caused by the absence of inflammatory components, as well as by the imprecise borders of these lesions. Chemoradiotherapy-induced OM can occur at any level of the oral mucosa, in particular in the soft palate, buccal floor, tongue, jugal mucosa, etc. (see Figure 2). However, if other locations are observed, such as the gingiva, dorsal part of the tongue, or lesions of the hard palate, other etiologies should be sought [22]. It has been found that the more keratinized oral mucosa is not usually affected by mucositis [26].

### 3.3. Tools Used to Assess OM

The lesions that characterize OM can be grouped into 3 categories: atrophic, erythematous and ulcerative. Ulcerations are accompanied by severe pain, colonization with different types of bacteria (with an increased risk of local and systemic infections), oral bleeding and compromise of physiological functions at the oropharyngeal level [27].

Different scales have been proposed for the assessment of OM which are particularly useful for clinicians dealing with this toxicity:The World Health Organization (WHO) scale evaluates OM as having 5 grades, from grade 0—normal mucosa, to grade 4 with deep lesions, when feeding the patient is not possible, making parenteral support necessary [28].The “Oral Assessment Guidelines” are used especially in pediatrics, where the degree of stomatitis and the condition of the oral cavity are assessed by inspecting the lips, oral commissures, tongue, the appearance of the oral mucosa membrane, saliva, gums, teeth, voice and swallowing process [29].The “Beck Assessment Scale”, in short BOAS, much similar to the previous one, is adapted, oral functionality being assessed with the help of local examination, registering scores from 5 to 20 [30].The “Oral Toxicity Scale” is used to evaluate the degree of oral stomatitis, similar to the previous ones. This instrument has been developed by Parulekar and uses symptomatic items for assessing the patients and sorting them by 5 grades [30].

Another particularly useful tool is that developed by the National Cancer Institute (NCI) Common Terminology Criteria for Adverse Events, in short, NCI-CTCAE, which is based on clinical examination, combined with functionality evaluation and local symptoms. In this regard, the oral mucosa is assessed as having 5 grades, starting with grade 1 with mucosal erythema, minimal symptoms, normal swallowing and unchanged diet, and going up to grade 4, in which advanced local necrosis is present, having significant spontaneous life-threatening bleeding with subsequent patient death (grade 5) [31].

The Radiation Oncology Group also assesses OM as 5 grades, from grade 0 to grade 5, where the oral mucosa shows necrosis and deep ulceration or significant bleeding, (See Table 1) [21,22,32]. 

These classification systems are extremely easy to apply by clinicians and particularly useful in the subsequent management of this toxicity. The side effects of OM should not be neglected: appearance of local pain, significant changes in the patient’s feeding process, leading in severe cases to the avoidance of oral route nutrition or drug administering, the patient needing parenteral nutrition or gastrostomy tubes, interruption of oncological treatment, an increased length of hospitalization, high doses of analgesics and opioids, all these having severe repercussions on the patient’s quality of life [21]. In a study on patients treated with chemotherapy for solid tumors, the cost per cycle of chemotherapy with OM was 9132 US dollars, as compared to 3893 US dollars per cycle of chemotherapy without OM [33]. OM has a profound impact on tumor response and long-term patient survival due to the necessity of unplanned breaks in cancer treatment, which subsequently leads to dose changes and increased recurrence with decreased survival rates [34].

### 3.4. Management of OM

In order to minimize major adverse effects of cancer treatment, it is necessary to reduce dosage or to even stop treatment completely. The management of oncological toxicities is a challenge for clinicians today, especially concerning the education of patients in regards to the oral hygiene, diet, and lifestyle changes, as well as the management of pain according to intensity, symptoms, prevention of complications, prophylaxis, or treatment of secondary infections [26]. A crucial aspect is an early diagnosis and appropriate treatment of dental diseases [9,35].

For the prevention and treatment of OM, various synthetic or natural substances are being sought, substances that could be effective, safe, easy to administer, and without side effects. In recent years, important clinical trials have been carried out and efforts have been made to introduce these substances into practice [34]. One of the highest authorities in the field is the Mucositis Study Group, of the Multinational Association for Supportive Care in Cancer, and the International Society of Oral Oncology (MASCC/ISOO) which first published the recommendations and guidelines based on multiple clinical trials. This group published in 2004 the first guideline based on 1197 publications, including recommendations for the management of OM, grouped into eight sections, seven of which refer to OM and one to gastrointestinal mucositis [36].

However, with the introduction of new cancer therapies, other types of mucositis associated with them have appeared, for example, mucositis secondary to immunotherapy or mucositis associated with targeted therapies. It is therefore necessary to update these guidelines and protocols based on new and effective research. In this respect, the seven sections on OM are kept, namely the importance of oral care before and during cancer therapy, the use of anti-inflammatory drugs, and the use of benzydamine mouthwash is recommended. Additionally of particular importance is photobiomodulation (applications using monochromatic light sources that have a cytoprotective effect or low-level laser therapy) based on recent but controversial evidence due to possible long-term carcinogenic effects. Used for its vasoconstrictor effect, cryotherapy limits the cytotoxic effects of cancer therapy. Studies also show the importance of growth factors and cytokines in preventing or treating OM. Additionally, antimicrobial agents, cover agents, anesthetics, and analgesics are recommended based on studies, namely morphine (topically administered), topical or systemic sucralfate, fluconazole, doxepin, and others. A special part of these recommendations is linked to natural agents such as glutamine, vitamin E, selenium, and honey for the prevention of OM. Other natural remedies and herbal combinations have not been included in this guideline due to insufficient clinical evidence [34,37].

These guidelines are based on studies that have taken into account criteria such as duration, the severity of the OM, and the pain caused by it. Although there are therefore numerous recommendations and various substances have been approved, as listed in the specific guidelines, an effective strategy for the prevention and treatment of OM has not yet been developed [34,36,37,38].

### 3.5. Evidence Regarding the Effectiveness of Bee Honey in Preventing and Treating Chemoradiotherapy-Induced OM

All of this research demonstrates the efficacy of bee honey in OM, and, as a result, it is recommended by researchers and included in international guidelines dealing with the prevention and treatment of this toxicity [37].

In a prospective, single-blind, randomized control trial conducted in India by Howdler et al., 40 patients were enrolled and divided into two groups, a study group, and a control group. Patients received two cycles of Taxol-based induction chemotherapy at 3-week intervals, then were radio treated concurrently with cisplatin-based chemotherapy 4 weeks after completion of induction chemotherapy. Patients from the study group slowly cleared their mouths with 20 mL of honey, after which they swallowed it slowly for 15 min, before and after treatment. In addition, they consumed a total of 100 mL of honey per day (1.2–1.5 mg/kgc/day) in divided doses, in order to maintain adequate serum antioxidant levels and to protect against oxidative stress. In terms of patient quality of life (QOL), there was a decrease in both groups (*p* < 0.05) up to 4 weeks, but post-therapy QOL increased significantly (*p* = 0.0001) and the mean improvement was better in the study group as compared to the control group. Thus, the study group (in comparison to the control group, who performed saline rinses) showed less impairment of swallowing function and less local pain, thus requiring less food restriction to liquid foods. The study concludes that honey is a simple, cheap, easy to administer, pleasant, and useful modality for the prevention and treatment of chemotherapy-induced OM [39].

In another prospective randomized study conducted in a tertiary hospital on 150 patients organized into three study groups, Mamgain RK et al. compared the efficacy of Ayurvedic preparation of Yashtimadhu with honey. Study group 3 used honey with local applications in the oral cavity as well as in a fixed quantity twice a day, compared to the other groups, which used the Ayurvedic preparation Yashtimadhu and conventional treatment against OM induced chemoradiotherapy (for head and neck cancer). In this study, 5% of the honey group developed grade 4 mucositis compared to 9.52% in group 1, where conventional treatment for mucositis was given. However, it was shown that the Ayurvedic preparation Yashtimadhu is superior to bee honey in accelerating the healing of mucositis and alleviating pain [40].

It has also been proven successful in preventing OM in a murine model study evaluating methotrexate-induced mucositis. A total of 24 albino rats were divided into four groups. OM was induced by intraperitoneal injection of 60 mg/kg of bodyweight methotrexate with the first signs of mucositis appearing from day 2 to day 5. Polyfloral, natural, unprocessed honey was used. The result of the study supports that bee honey is an effective agent for the improvement of chemotherapy-induced OM, by decreasing inflammation, as compared to the control group [41].

Noam Yarom et al., along with the Mucositis Study Group, part of the Multinational Association of Supportive Care in Cancer/International Society of Oral Oncology (MASCC/ISOO) using 29 studies in Part I and 49 studies in Part II, have recommended in this guideline the topical or systemic administration of bee honey for the prevention or treatment of OM, as well as for decreasing the severity of this oncological toxicity. In this study, locally and systemically combined honey is used: “natural” royal jelly honey, honey extracted from Camellia sinensis, Thymus, and Astragale, from the Western Ghats forests, from Trifolium alexandrenum, or unspecified [42].

In research done by Karsten Münstedt et al. which reviewed 17 randomized studies on the use of honey in chemoradiotherapy-induced OM from 2000 to 2018, they compared the benefits of conventional honey against Manuka honey. The focus was on the high amount of methylglyoxal in Manuka honey, a cytotoxic substance that can alter proteins, including DNA, causing tissue dysfunction, aging, and disease, and can also delay the healing process of lesions. The study therefore recommends the use of conventional honey in the prevention of oral mucositis [19].

On the other hand, Hunter et al. highlight the antioxidant, antibacterial and anti-inflammatory properties of honey in a paper that includes 13 studies in which honey is applied topically to patients undergoing chemoradiotherapy. It was highlighted that honey reduced the severity or duration of the disease as compared to control groups (*p* < 0.05) [11].

In 63 studies on head and neck surgery, Theresa Tharakan et al. demonstrated the efficacy of honey in preventing and treating OM in ENT oncology patients, in pain control after tonsillectomies, or in the treatment of upper respiratory tract infections. The study lists the type of honey used: thyme honey, polyfloral honey, Ziziphus honey, pure or a diluted honey oral rinse. The study also provides evidence in favor of honey in the treatment of chronic diseases such as rhinosinusitis, allergic and fungal diseases, or in the healing process of lesions associated with hearing aids or otitis externa [43].

In his meta-analysis, Chao Yang analyzed 17 randomized studies involving 1265 patients and 13 groups. In the honey-treated group, the therapeutic effect of honey in treating moderate–severe chemoradiotherapy-induced OM, was followed. Pure natural honey, according to this study, is therapeutically superior and decreased the onset time of OM (OR 0.41, CI = 0.08–0.73); no increase in adverse effects was observed in the study. Therefore, honey can be recommended as a first-line adjuvant therapeutic agent in the treatment of OM. This evidence supports the fact that honey accelerates tissue repair and healing from chemoradiotherapy-induced oral mucosal lesions [44].

In a meta-analysis comprising 19 randomized controlled trials involving 1276 patients, Tzu-Ming Liu et al. observed that the application of honey reduces the extent of radiochemotherapy-induced oral mucositis. The action of honey was also observed in the prophylactic phase, where a group receiving honey had registered an RR of 0.18, with a 95% confidence interval, as follows: CI = 0.09 to 0.41; in the treatment phase, patients given honey registered a significant pain score in month 1 of treatment, having a weighted mean difference of WMD = −3.25, 95% CI = −4.41 to −2.09; at the end of treatment, the following values were recorded: WMD = −2.32, 95%, with CI = −4.47 to −0.18.

The outcome of the studies is also favorable in terms of decreased incidence of intolerable mucositis in the honey-treated group, with a RR of 0.48 (95% CI = 0.26 to 0.87) [45].

In a meta-analysis comprising 28 randomized trials on 1861 patients, Ya Ying-Yu et al. established a ranking of agents used in the prevention and treatment of OM. Thus, the results of the study showed that chlorhexidine, benzidamine, honey, and curcumin were more effective than placebo solutions (*p* < 0.05), and honey and curcumin were more effective than povidone-iodine (*p* < 0.05). The study provides an important theoretical basis and indicates that curcumin and honey can be recommended as viable options for the prevention of OM [46].

One of the most comprehensive studies on this issue was carried out by MASCC/ISOO, a group that regularly updates guidelines on the management of mucositis. The guidelines were drafted based on 1197 publications dated between 2011 and 2016, also evaluating randomized clinical trials published up to 2019. The MASCC/ISOO guidelines recommend, among other agents, bee honey in the natural agents section. These guidelines are the most internationally used guidelines considered effective interventions based on strong evidence and extensive research conducted in order to manage this non-hematological toxicity post chemoradiotherapy [37].

The efficacy of bee honey was also compared with agents other than chlorhexidine and curcumin. In a large analysis conducted by Xu Zhang and co-workers, which involved 36 studies and registered a total of 2594 patients, they compared 10 mouthwashes, the data being included in the Bayesian network analysis. Thus, according to this analysis, mouthwash with honey (with an odds ratio [OR] of 0.17, 95% and a CI 0.09 to 0.30), chamomile (with an OR of 0.09, 95% and a CI 0.01 to 0.52), curcumin (with an OR of 0.23, 95% and a CI of 0.08 to 0.67), and benzidamine (having an OR of 0.26, 95% with a CI of 0.12 to 0.54) registered superior results as compared to placebo. In turn, honey mouthwash proved to have higher efficacy than mouthwash with chlorhexidine (registering an OR of 0.34, 95%, with a CI of 0.12 to 0.92), sucralfate (with an OR of 0.26, 95%, and a CI of 0.06 to 0.96), or povidone-iodine (with an OR of 0.30, 95% with a CI of 0.11 to 0.82).

This study therefore concludes that honey, chamomile, curcumin, and benzidamine are the most advantageous in the prevention of chemoradiotherapy-induced OM [47], as well as other remedies obtained from natural extracts and used in various cutaneo-mucous pathologies [48].

The results of the present study on the benefits of bee honey in patients with head and neck cancer undergoing radiochemotherapy are summarized in Table 2 below.

### 3.6. Other Benefits of Honey

Honey has proven its efficacy against chemotherapy-induced stomatitis in a group of patients with colon cancer, knowing that this neoplasm requires much higher doses of chemotherapy. The study was conducted in the city of Minia, on a group of 60 patients divided into two groups, one control and one study group. The study confirms the efficacy of honey in the patients of the study group who recorded an improvement in oral function, thus obtaining a reduction in the severity of oral stomatitis [30].

Regarding pediatric patients with neoplasms who developed oral mucositis after chemo-radiotherapeutic treatment, there are few studies with promising results. One such study was conducted in Saudi Arabia and it was published in 2017. This randomized controlled trial was conducted on 40 pediatric patients with hematological cancer who were treated with chemoradiotherapy and who developed OM. Patients from the control group performed honey rinses followed by rinsing the oral cavity with saline 3.4 times daily. The results of the study are surprising, showing a reduction of grade III and IV OM, and concerning post-radiotherapy toxicity, the study recorded a decrease of more than 80% of radiotherapy-induced OM. It is therefore concluded that honey, being a natural, inexpensive product with few side effects, is well tolerated by patients, especially pediatric ones, has a pleasant taste, and is easy to administer. It is recommended to use topically as part of the standard care in chemoradiotherapy-induced OM [49].

The study also investigated the impact of bee honey on pediatric patients admitted to the pediatric intensive care unit of a university hospital in Turkey, who developed OM. The study involved a group of 150 patients, randomly divided into six groups, who were administered chlorhexidine, vitamin E, and bee honey. At the end of the study vitamin E was found to be the most effective solution in the management of OM, followed by honey and chlorhexidine [38].

## 4. Discussion

Currently, chemoradiotherapy-induced OM benefits from a multitude of pharmacological and non-pharmacological therapeutic options. Although numerous agents exposed in the present study are available, researchers have not yet reached a consensus regarding the prevention and treatment of this toxicity [50]. There are many studies in the literature on the usefulness of honey in the prevention and treatment of OM based on the biological properties of this agent. As an anti-inflammatory and antimicrobial agent and immunomodulator, this natural product has a great impact on OM [8,9,10,11,13,51,52]. For example, Pradip Kumar Maiti et al. conducted a study on 50 patients diagnosed with head and neck cancer who were radiotreated and divided into two study groups. The study arm was given 20 mL of honey 15 min before, 15 min after treatment, and a similar amount at bedtime. After evaluation, there was a significant reduction in grade 3 or 4 OM in the honey arm from 18% to 41% in the control arm [53]. The working method is similar to Howdler et al.’s, mentioned in this paper. Both Pradip Kumar Maiti et al. and Howdler conclude that honey is an inexpensive, readily available, effective agent for treating OM. The difference between these studies is in the oncological treatment applied to the patients, i.e., radiotherapy alone versus radiochemotherapy in combination with cisplatin-based chemotherapy. Not all studies specify the type of honey used. Howdler et al. used 20 mg of fresh, organic, unprocessed honey and in the study by Pradip Kumar Maiti et al. honey of unspecified origin was used but the recommendations of the studies are similar [39,53]. On the other hand, Motallebnejad M et al. conducted a double-blind randomized clinical trial in which 40 patients randomly assigned to two groups are examined. The study protocol is similar to the studies described above. However, the type of honey used is specified, this being pure natural honey obtained mainly from Thymus and Astragale in the Alborz Mountains, northern Iran. It is assumed that the beneficial effect of honey is based not only on its antibacterial properties but also on its geographical location, pollen source, season, type of bee, or other factors influencing its quality [10,11,54]. In contrast in the subspecialty review of Tharakan, T et al. [43], in which oncology patients are administered thyme honey, polyfloral honey, Ziziphus honey, pure or diluted honey oral rinse with satisfactory results, Karsten Münstedt [19] evaluated the efficacy of conventional bee honey and Manuka honey on radiochemotherapy-induced OM. Conventional honey has been shown to be more beneficial than Manuka honey, which requires caution when administered [11] for example due to adverse effects (severe nausea, vomiting, and severe burning sensations in the mouth) it was necessary to change the protocol in a study conducted by Parsonset et al. [11,54,55]. On the other hand, Hunter et al.’s study also found adverse effects of Manuka honey in a group treated with this type of honey [11] compared to Bardy et al., who conducted a double-blind, placebo-controlled, randomized trial with 131 patients. Consequently, a different form of honey presentation is recommended using a more liquid formula with active ingredients [11,56]. In other respects in the systematic reviews and meta-analyses [37,42,44,45] introduced in the present study, reference is made to honey combined locally and systemically, “natural” honey, royal jelly, honey extracted from Camellia sinensis, Thymus, and Astragale, from the Western Ghats forests, from Trifolium alexandrenum, pure natural honey, manuka, local honey, or unspecified honey applied locally or administered systemically. The results of studies are positive, honey administered either in dilution or in pure form has beneficial effects on OM [43,46,47]. Although the literature, as we have extensively exposed in this study, supports the benefits of bee honey in preventing and treating OM in patients with head and neck cancer, this topic is still controversial. Effective management of this toxicity is needed, but in neoplastic patients who are given different types of treatments such as pain relievers and anti-inflammatory drugs, which may influence the results of the research, these studies sometimes seem inconsistent [44]. On the other hand, it is very difficult to control the purity of honey, as it differs depending on a multitude of factors. In order to avoid inconsistent results in future research, it is necessary to specify and recognize the differences between different types of honey or to identify the active substances of this agent responsible for its beneficial effect on OM [19]. Despite these pros and cons, there is sufficient evidence and scientific studies conducted by researchers recommending bee honey to be included in the guidelines responsible for the management of this post-radiochemotherapy toxicity [34,36,37,38,42].

Despite its many benefits, honey consumption is still limited. Many researchers also warn patients about potential health problems caused by the excessive consumption of honey. Although honey has extraordinary anti-inflammatory, antibacterial, antiviral, anticancer, and tissue-repairing properties, caution is still required in its consumption [52]. Moreover, honey is a food subject to adulteration. In its composition there must not be any other ingredient, flavor, or foreign substance. Honey must not be overheated in order not to lose its properties. Pure honey can be altered either directly by adding adulterants, indirectly by feeding bees, or by mixing honey with other low-quality products [57]. Apart from the nutrients and compounds that give it these beneficial physicochemical properties, there may be contaminants in the composition of honey, such as pesticides, antibiotics, heavy metals, or other toxic agents, caused by exposure to the environment, faulty handling by beekeepers, or even the presence of numerous bee diseases that can ultimately affect the quality of honey. Honey can be contaminated by various chemicals or pathogens, which is why it is recommended that before consumption honey should be sterilized in order to remove contaminated agents without losing the therapeutic properties of honey. The use of honey should not be recommended for cancer patients with associated diabetes.

Another important problem is the presence of allergens that can cause severe anaphylactic reactions. Honey can also be a factor in the development of dental caries, therefore careful care of the oral cavity is recommended, especially during antineoplastic treatment.

Given the benefits but also the contraindications, manufacturers are encouraged to label each beekeeping product appropriately, especially honey [52].

## 5. Conclusions

OM caused by oncological treatments is an important clinical problem that does affect the head and neck cancer patients’ life quality. Honey has a special nutritional and therapeutic value. Due to the numerous physiochemical properties conferred in general by polyphenols, they determine pharmacological and preventive actions exposed in this paper. Although the way honey acts is not completely known, various studies and extensive analyses conducted by researchers certify honey as a successful natural agent in the prevention and treatment of oral mucositis induced by chemoradiotherapy.

Therefore, honey can be recommended and is included in the international guidelines that manage this non-hematological toxicity, namely the MASCC/ISOO guidelines.

However, new studies and more rigorous evidence on the action of different compounds of this natural product on oral mucositis are still needed, knowing that honey is greatly influenced by the environment, the type of bees, the way of processing, etc. Therefore this topic is still controversial. This line of research has the potential to open up new opportunities in palliative symptomatic treatment in other cancers, given previous research on the modification of epigenetic pathways and in cell cycle alteration in patients with different types of carcinomas and other sites [58]. The present study is limited because it covers a short period of time. Also some studies in the references are applied on a limited number of patients or are carried out in single study centres. Although reference is made to various types of honey, large studies are needed covering more geographical areas worldwide or to study other specific types of honey (Acacia, Mentha piperita...) considering their recognized properties [59,60,61,62]. In the present context, taking into account the recommendations of studies and international guidelines, the authors support the use of bee honey in the prevention and treatment of OM knowing that this natural agent is easy to administer, effective, highly addressable, inexpensive and easy to purchase, with multiple benefits but especially without side effects.

## Figures and Tables

**Figure 1 medicina-58-00751-f001:**
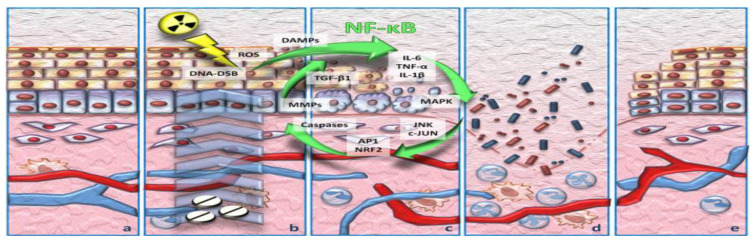
(after Claudio Pulito-2020) [17]. Pathobiology of oral mucositis: (**a**) normal tissue; (**b**) first phase and initial injury response, (**c**) injury signal boost, (**d**) ulceration, (**e**) tissue re-epithelialization.

**Figure 2 medicina-58-00751-f002:**
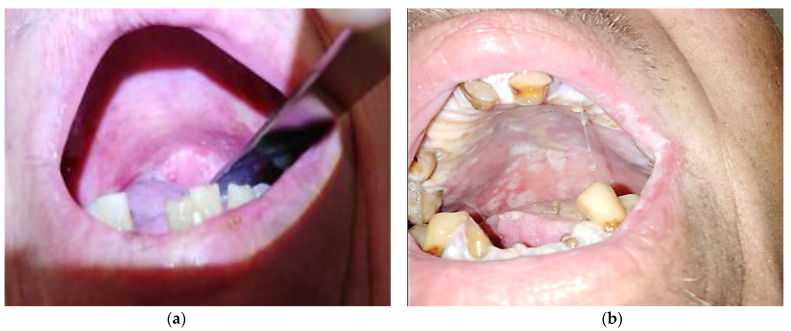
Oral mucositis in patients with head and neck cancer treated with concurrent radiochemotherapy. (**a**) Erythema, (**b**) ulcerations covered by fibrinous pseudomembranes and secondary fungal colonization (authors’ collection May 2022).

**Table 1 medicina-58-00751-t001:** Examples of tools used to assess OM.

	Grade 0	Grade 1	Grade 2	Grade 3	Grade 4
WHO		erythema and soreness	ulcers, able to eat solids	ulcers, requires a liquid diet (due to mucositis)	ulcers, alimentation not possible (due to mucositis)
RTOG		(mild) irritation, mild pain, does not necessarily require analgesics	(moderate) patchy mucositis with inflammation and serosanguinal secretions, moderate pain may be present requiring analgesics	(severe) confluent or fibrinous stage of mucositis, with severe pain requiring narcotics	(life threatening) deep ulceration, bleeding, or necrosis
OMAS Erythema Ulceration	Normal	<1 cm^2^ Not severe	1–3 cm^2^ Severe	>3 cm^2^	
NCI-CTCAE		erythema, painless ulcers or mild pain in the absence of lesions	edema, painful erythema, and ulcers, but patients may eat or swallow	severe ulcers present, the patient requires enteral/parenteral nutrition or prophylactic intubation	death caused by this toxicity

Abbreviations: WHO: World Health Organization; RTOG: The Radiation Oncology Group; OMAS: Oral Mucositis Assessment Scale; NCI-CTCAE: National Cancer Institute (NCI) with common terminology criteria for adverse events.

**Table 2 medicina-58-00751-t002:** Evidence regarding the effectiveness of bee honey in preventing and treating chemoradiotherapy-induced OM.

Article Type; Authors	Number of Patients or Studies Used	Oncological Treatment	Objective	Type of Honey and How to Use It in the Study Group	Substances Used in the Control Group	Results
1. Prospective single-blind randomized control study; Howlader, D. et al. [39]	40 patients divided into 2 arms	radiochemotherapy together with cisplatin-based chemotherapy 4 weeks after completion of induction chemotherapy	to assess clinical benefits and improve quality of life in patients with head and neck cancers after honey administration	− raw, organic, unprocessed honey − mouthwash with 20 mL honey 15 min before radiation exposure and 15 min after exposure and 6 h after radiotherapy, a total of 100 mL (1.2–1.5 g/kg) of honey per day in divided doses	− mouth rinsing with normal saline 15 min before radiation exposure and 15 min after radiation exposure	− the study group showed less impairment of swallowing function, less local pain − QOLdecreases in both arms (*p* < 0.05) up to 4 weeks − QOL after therapy increased significantly (*p* = 0.0001) in the study group − mean improvement was better in the study arm compared to the control arm − bee honey is a simple, cheap, easy to administer, pleasant, and useful modality in the prevention and treatment of oral mucositis induced by chemoradiotherapy
2. Prospective randomized control study Mamgain, R. K. et al. [40]	150 patients initially enrolled randomly assigned to 3 arms	local EBRT at 6 MV LINAC by conventional fractionation, average dose = 60 Gy × 5 days/week 6 weeks concurrent with cisplatin	to evaluate the efficacy of Ayurvedic preparation in oral mucositis in head and neck cancer patients receiving concurrent chemoradiotherapy	− unspecified honey − group III (40 patients): honey applied locally + 1 teaspoon × 2/day orally	− group I (40 patients): treated with conventional mucositis drugs − group II (45 patients): mouthwash with warm salted water, then paste 5 g of Yashtimadhu powder mixed with honey, topical application in the oral cavity × 2/day + 500 mg of Yashtimadhu capsule × 2/day orally	− 20% of patients in the honey group developed oral mucositis grade 3 compared to 15.5% in the Yastimadhu study group − reduced hospitalization in patients with Ayurvedic preparation administration compared to other patient groups
3. Murine model study Khanal, L. et al. [41]	24 albino rats randomly assigned to 4 working arms	intraperitoneal methotrexate at a dose of 60 mg/kg	to demonstrate the efficacy of bee honey on chemotherapy-induced oral mucositis	− polyfloral honey, natural, unprocessed − group IV: treated with honey methotrexate	− group I: treated with normal saline honey − group II: treated with distilled water-saline solution − group III: treated with distilled water methotrexate	− bee honey is an effective agent for the relief of chemotherapy-induced OM by decreasing inflammation compared to the control group − restoration of chemotherapy-induced body weight in the honey group
4. Systematic review Yarom, N. et al. [42]	78 papers: 49 were included in this review +9 publications reported in the previous update of the guidelines describing 26 different interventions falling within the honey field	radiotherapy with or without chemotherapy in patients with head cancer pediatric patients with hematological or solid cancers treated with chemotherapy	to update the clinical practice guidelines for OM management that have been developed by MASCC/ISOO. This part focuses on honey, herbal compounds, saliva stimulants, probiotics, and miscellaneous agents	− honey combined locally and systemically: “natural” honey, royal jelly, honey extracted from Camellia sinensis, Thymus, and Astragale, from the Western Ghats forests, from Trifolium alexandrenum, or unspecified	− benzidamine and nystatin − topical lidocaine − mixture of − honey and caffeine and steroids (8 mg − betamethasone)	− MASCC/ISOO guideline update suggests application of honey, combined topically and systemically, for prevention of OM in H&N cancer patients treated with either RT or RT-CT
5. Systematic review Münstedt, K. et al. [19]	17 randomized trials	radiotherapy or radiotherapy with combined chemotherapy	to evaluate the efficacy of conventional bee honey or Manuka honey on radiochemotherapy-induced OM	− conventional honey and Manuka honey	− with saline 0.9% − povidone-iodine − with water chamomile − with salt soda and − benzydamine gargle − with placebo gel − benzydaminz hydrochloride − with golden syrup − with lignocaine gel − standard care	− Manuka honey can delay the healing process − studies recommend the use of conventional honey in the prevention of OM
6. Systematic review Hunter, M. et al. [11]	13 randomized controlled trials with 634 patients	chemotherapy or radiotherapy	to demonstrate the efficacy of bee honey on oral mucositis induced by chemotherapy or radiotherapy	− undiluted topically applied honey from any floral source and Manuka honey	− placebo treatment − standard routine oral care − saline rinses with different concentrations (0.9%, 0.09%, unspecified) − anesthetic and analgesic solutions (7.5% benzocaine gel, 15% bendamdamine hydrochloride, lignocaine gel) − placebo gel − own mixture	− honey reduced the severity and duration of OM compared to control groups (*p* < 0.05) − one group treated with Manuka honey (*n* = 1) with adverse effects
7. A subspecialty review Tharakan, T. et al. [43]	13 randomized controlled trials with 634 patients	chemotherapy or radiotherapy	to demonstrate the efficacy of bee honey on oral mucositis induced by chemotherapy or radiotherapy	− thyme honey, polyfloral honey, Ziziphus honey, pure or diluted honey oral rinse	− no treatment − topical lidocaine − gold syrup − sugar cane syrup (positive control) − povidone-iodine rinse − placebo − mouthwash with chamomile − benzidamine and soda with salt alternate rinses − betamethasone PO − solution	− more effective treatment in patients treated with radiotherapy alone than those with radiochemotherapy − the honey delayed the onset of OM − decreased the number of treatment interruptions in honey study arms − regulated body weight in honey patients
8. A systematic review and network meta-analysis Yang, C. et al. [44]	17 studies involving 1265 patients grouped into 13 arms	chemotherapy or radiotherapy	to demonstrate the efficacy of bee honey on oral mucositis induced by chemotherapy or radiotherapy	− pure natural, − Manuka or topical honey − local honey	− placebo − regular care − benzocaine − benzidamine − caramel dye − chamomile − Golden syrup − lidocaine and other	− honey treatment increased the therapeutic effects of treatment from 0.25, 0.14 to 0.46 − pure honey is therapeutically superior from 0.05, 0.01 to 0.46 − the therapeutic effect of honey in the treatment of moderate-severe OM induced by chemotherapy is observed − reduces the time of onset of OM (OR 0.41, CI = 0.08–0.73)
9. A meta-analysis of randomized controlled trials Liu, Tzu-Ming et al. [45]	19 randomized controlled trials with 1276 patients	radiochemotherapy	reduction of OM	− pure natural or Manuka honey	− the same protocol as the group treated with honey except that the honey was not used − placebo − lidocaine − glycerin − anesthetic and antacid solution − golden syrup − sugar-free placebo gel − benzidamine 0.15% hydrochloride − normal saline 0.9% − 1 mL betadine and 100 mL water = mouthwash	− honey reduced the development of OM in the prophylactic phase RR = 0.18, 95% confidence interval CI = 0.09 to 0.41 − significant decrease in pain scores in the first month of treatment (weighted mean difference WMD = −3.25, 95% CI = −4.41 to −2.09) and at the end of treatment (WMD = −2.32, 95% CI = −4.47 to −0.18) − honey is recommended during and after radiochemotherapy to prevent and treat OM
10. A network meta-analysis of randomized controlled trials Ya-Ying Yu et al. [46]	28 randomized controlled trials with 1861 patients	radiochemotherapy	prevention and treatment of OM, evaluate the effect of different oral care solutions	− unspecified honey	− placebo − various oral care solutions − chlorhexidine − benzidamine − curcumin − povidone-iodine − alopurinol − sucralfate − GM-CSF − aloe	− chlorhexidine, benzidamine, honey, and curcumin were more effective than placebo (*p* < 0.05) − honey and curcumin were more effective than povidone-iodine (*p* < 0.05) − important theoretical evidence indicating that curcumin and honey may be recommended for the prevention of OM
11. New systematic review and update the clinical guidelines Sharon Elad et al. [37]	1197 randomized controlled trials with 1861 patients	radiochemotherapy	prevention and treatment of OM	− unspecified honey applied topically and administered systemically	− oral care protocols combined with agents − sodium bicarbonate solution − benzidamine mouthwash − topical morphine mouthwash 0.2% (pain associated) − sucralfate − oral glutamine − saline solution − placebo − morphine (topical), sucralfate (topical/systemic), fluconazole (systemic), miconazole (topical and systemic), − mucoadhesive hydrogel, polyvinylpyrrolidone, doxepin, fentanyl (transdermal) − natural agents: vitamins, minerals, and nutritional supplements on OM, − including glutamine, elemental diet, zinc, calcium phosphate, vitamin E, selenium, folinic acid − calcitriol	− guidelines containing recommendations for the management of OM grouped into 7 sections for OM and 1 for gastrointestinal mucositis are developed − the MASCC/ISOO guideline recommends among the agents that prevent or treat OM and bee honey in the section natural agents in H&N cancer patients receiving treatment with either RT or RT-CT
12. A Bayesiannetwork analysis Xu Zhang et al. [12]	36 randomized controlled trials with 2594 patients	radiochemotherapy, total radiation dose =50 Gy	to compare the preventive effect of ten mouthwashes in intolerable OM	− mouthwash with honey as common reference	− variety of mouthwashes: − aloe vera − benzidamine − chamomile − chlorhexidine − curcumin − lactobacillus brevis − Na bicarbonate − povidone-iodine − succralfate mouthwash	− compares 10 mouthwashes: − Bayesian network analysis showed that mouthwash with honey (odds ratio OR 0.17, 95% CI 0.09 to 0.30), chamomile (OR 0.09, 95% CI 0.01 to 0.52), curcumin (OR 0.23, 95% CI 0.08 to 0.67) and benzidamine (OR 0.26, 95% CI 0.12 to 0.54) were superior to placebo − mouthwash with honey was more effective than mouthwash with chlorhexidine (OR 0.34, 95% CI 0.12 to 0.92), sucralfate (OR 0.26, 95% CI 0.06 to 0.96), and povidone-iodine (OR 0.30, 95% CI 0.11 to 0.82) − according to rank probabilities, chamomile, honey, curcumin, and benzidamine are the most advantageous in the prevention of chemoradiotherapy-induced OM − easy agent to procure, administer, and acceptable, especially in pediatrics

## Data Availability

Not applicable.

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
