# Peer review of "Oral Mucositis Induced by Chemoradiotherapy in Head and Neck Cancer—A Short Review about the Therapeutic Management and the Benefits of Bee Honey"

_medicina, 2022, doi:10.3390/medicina58060751_

Round 1
Reviewer 1 Report
- Please support each statement of fact with a Reference citation.
- Please describe the current state of knowledge on the treatment of oral mucositis with special emphasis on their outcome (please indicate which are used in clinical practice and which are only the subject of the research).
Reviewer 2 Report
Dear Authors,
Highly up to date review summarizing most recent clinical trails with use of honey in treatment of oral mucositis (OM) induced by oncological treatment.
There is a great importance in meta-analyzing clinical trials to bring honey therapy into clinical practice. More importantly, it is a type of honey need to be distinguished. Not all honeys have the same affect. It is important that some honeys might be more suitable for treatment particular health issues while other don't.
Article is nicely written and important. Even though, there are some areas that need to be improved.
Major Issues
Do add table where all mentioned trials are summarized such as it is presented in Münstedt et al., 2019 (Ref.; Design; Sample, intervention and control group; Oncological treatment; Endpoints; Main results (Only significant results))
Line 242 - “Although there are numerous recommendations and various substances have been approved, an effective strategy for preventing and treating OM has not yet been developed [27]” What are the substances that authors are mentioning? Name with citations.
Dicsussion needs to be re-written. The section with general information on honey should be moved to introduction and picked only relevant information. With focus on wound healing aspects and those who might play important role in OM treatment. Do preceive a difference among types of honey.
Section “4.2. Other benefits of honey” should be moved to Results. Also no need to mentioned those clinical trails where honey was used to treat other health issues. Make OM priority.
In discussion, collected data and their relevance must be discussed. As well as to bring various hypothesis why is it/ is it not honey better option; in which cases and why it is so. Discuss those biological activities which might have the biggest impact on OM. Is the wound healing, immunomudulatory and anti-inflammatory affect together with antibacterial? (With references)
Based on collected data is there a honey type that comes better in treatment of MO than others? Try to propose some criteria which must honey for MO treatment meets.
Minor issues
Other points:
Search strategy items needs to be mentioned in Methods.
Line 20 - missing Romania
Line 165 - See Figure 1
Line 169 - move to Figure 1. description
Table 1. - smaller font size
Line 228 - missing citation
Line 358 - no need to full text the abbreviation, because it is already explain previously in Line 309
Reviewer 3 Report
The manuscript authored by Daniela Jicman et al. is well-written, presenting the ability of bee honey to be used in the prevention and treatment of Oral mucositis caused by Chemoradiotherapy in Head and Neck Cancer.
1. The Oral mucositis should be written in full once, and then abbreviated throughout the
manuscript as OM. Authors keep on writing Oral mucositis in full even after
abbreviated, e.g., on line 91, 103, 135, etc.
2. Bee honey should be briefly presented in the introduction section as a potential
bioactive agent for the prevention and treatment of Oral mucositis.
3. Authors should add one or two figures in the manuscript for it to be well presented.
Regards
Round 2
Reviewer 2 Report
The manuscript has been approved. But the formating is not unified and disturbing. Text must be aligned. The table should contain an overview of the most important and significant results and serves as a tool for easier orientation. Therefore, I recommend some additional changes:
- Reference for the article in the first column.
- Control, which was used in separate column.
- Results - only significant results and main outcome of the trial.
- Reduce space between the lines to make the text more dense.
- Fit the table in the page. Also you can divide the table in two parts if needed.
Discussion - dense text. No need the space between lines.
Discussion does not contain sub paragraphs. Include 4.1
Line 195 - remove
Line 198-200 - re-formate the figure description
Line 262 - remove
Line 459 - space in front of “Although”.
Correct reference "et all" change to "et al." how it should be written correctly.
Correct backspaces following coma. The whole formate of the paper is disturbing and not unite. It does not look like final version at all.
English spell check needs to be done.
